# Antioxidant Effect of Nanoparticles Composed of Zein and Orange (*Citrus sinensis*) Extract Obtained by Ultrasound-Assisted Extraction

**DOI:** 10.3390/ma15144838

**Published:** 2022-07-12

**Authors:** Ana G. Luque-Alcaraz, Miranda Velazquez-Antillón, Cynthia N. Hernández-Téllez, Abril Z. Graciano-Verdugo, Nadia García-Flores, Jorge L. Iriqui-Razcón, María Irene Silvas-García, Aldo Zazueta-Raynaud, María J. Moreno-Vásquez, Pedro A. Hernández-Abril

**Affiliations:** 1Ingeniería Biomédica, Universidad Estatal de Sonora, Hermosillo 83100, Mexico; ana.luque@ues.mx (A.G.L.-A.); mirandavelazquez1999@gmail.com (M.V.-A.); cynthia.hernandez@ues.mx (C.N.H.-T.); jorge.iriqui@ues.mx (J.L.I.-R.); aldo.zazueta@ues.mx (A.Z.-R.); 2Departamento de Ciencias Químico Biológicas, Universidad de Sonora, Hermosillo 83000, Mexico; abril.graciano@unison.mx (A.Z.G.-V.); mariadejesus.moreno@unison.mx (M.J.M.-V.); 3Departamento de Física, Universidad de Sonora, Hermosillo 83000, Mexico; nadian.garciafl@gmail.com; 4Departamento de Investigación y Posgrado en Alimentos, Universidad de Sonora, Blvd. Luis Encinas y Rosales S/N, Col. Centro, Hermosillo 83000, Mexico; maria.silvas@unison.mx

**Keywords:** zein nanoparticles, ultrasound-assisted extraction, antioxidant capacity

## Abstract

In the present research, an orange extract (OE) was obtained and encapsulated in a zein matrix for its subsequent physicochemical characterization and evaluation of its antioxidant capacity. The OE consists of phenolic compounds and flavonoids extracted from orange peel (*Citrus sinensis*) by ultrasound-assisted extraction (UAE). The results obtained by dynamic light scattering (DLS) and scanning electron microscopy (SEM) indicated that zein nanoparticles with orange extract (NpZOE) presented a nanometric size and spherical shape, presenting a hydrodynamic diameter of 159.26 ± 5.96 nm. Furthermore, ζ-potential evolution and Fourier transform infrared spectroscopy (FTIR) techniques were used to evaluate the interaction between zein and OE. Regarding antioxidant activity, ABTS and DPPH assays indicated no significant differences at high concentrations of orange peel extract and NpZOE; however, NpZOE was more effective at low concentrations. Although this indicates that ultrasonication as an extraction method effectively obtains the phenolic compounds present in orange peels, the nanoprecipitation method under the conditions used allowed us to obtain particles in the nanometric range with positive ζ-potential. On the other hand, the antioxidant capacity analysis indicated a high antioxidant capacity of both OE and the NpZOE. This study presents the possibility of obtaining orange extracts by ultrasound and coupling them to zein-based nanoparticulate systems to be applied as biomedical materials functionalized with antioxidant substances of pharmaceutical utility.

## 1. Introduction

Reactive oxygen species (ROS) and reactive nitrogen species (RNS) readily attack a variety of critical biological molecules such as lipids, DNA, and essential cellular proteins, causing alterations in the normal physiology of cells and organs, activating and accelerating disease processes [1]. Several studies show adverse effects caused by oxidative stress; an imbalance between ROS generation and the antioxidant system alters gene expression and cell signaling, causing cancer progression or death [2]. Another ROS-related disease is cerebral ischemic stroke, a leading cause of death and disability in humans [3]. It has recently been pointed out that oxidative stress may be a significant factor in COVID-19 pathogenesis due to its role in the response to infections [4]. In addition, oxidative stress transforms 3CLpro (a key proteinase for SARS-CoV-2 replication and a significant target for antiviral drug development) into an insoluble and enzymatically active form, leading to increased viral replication/transcription, suggesting the therapeutic potential of antioxidants in the clinical treatment of COVID-19 patients [5]. Antioxidants are compounds used by the body to eliminate free radicals, which are highly reactive chemicals that introduce oxygen into the cells and cause oxidation in different cell parts, changes in the DNA, and various changes that accelerate the body’s aging [6]. However, the body’s antioxidant defenses are sometimes not entirely efficient, and there is an increase in the formation of free radicals, i.e., oxidative stress. In cases such as these, some authors mention the importance of consuming phenolic compounds such as flavonoids [7] and highlight the importance of exogenous consumption of antioxidants through diet to further protect the body and help fight heart disease and cancer [8].

The use of organic solvents generates waste, and the existence of solvent traces in the extracts obtained is also possible [9] However, recently, some studies have been published on the extraction of bioactive molecules using non-polluting technologies, including supercritical fluids—mainly supercritical state carbon dioxide [10] This methodology allows for obtaining high-quality extracts and simultaneously doing it in an environmentally friendly way [11]. Supercritical CO_2_ extraction allows for selective extraction, uses moderate temperatures, allows for easy separation of the compounds obtained, does not present toxic residues in the extracts, and is an environmentally friendly technology that allows recycling of the solvents used. A recent technology is the use of UAE, offering some valuable advantages such as higher productivity, yield, and selectivity; shorter process time; high quality in the final product; and the reduction of chemical/physical risks. It is also more environmentally friendly as it does not generate hazardous industrial waste [12]. The extraction is based on the application of ultrasound (20–100 kHz) which causes the implosion of cell cavitation bubbles in which acoustic waves propagate, leading to the disruption of cell membranes. This action facilitates the penetration of the solvent into the cells, thus enhancing mass transfer and the release of bioactive compounds. Ultrasonication is a method for obtaining extracts and uses physical and chemical phenomena that are fundamentally different from those conventionally applied in extraction, processing, and preservation techniques [13]. The compounds obtained in many cases present low solubility and the low stability of phenolic phytochemicals in the gastrointestinal tract, which plays an essential role in their low absorption rate. Among these bioactive molecules, some flavonoids have been used—mainly those extracted from citrus peel—demonstrating applications in the biomedical area by exhibiting antioxidant, antihypertensive, anti-inflammatory, and anticarcinogenic capacity through different mechanisms [5]. Their affinity determines the aqueous solubility of phenolic phytochemicals with water molecules, and molecular weight plays a crucial role in aqueous solubility. There are alternatives to reduce this problem. The synthesis of polymeric nanoparticles has been proposed to give them the property of integrating bioactive molecules and protecting them from degradation in their physiological distribution route to meet their objective of reaching the critical area of action [14]. The interest in the study of micro- and nano-systems developed based on chitosan has attracted significant attention in applications related to the encapsulation of bioactive compounds due to its recognition as safe (GRAS) and its biological properties such as biodegradability, biocompatibility, and low or no toxicity, as well as its ability to form films, membranes, gels, beads, fibers, and micro- and nanoparticles [15].

In recent years we have witnessed increasing research in the field of nanotechnology. In particular, the boom in polymeric nanoparticles, used in various areas of life and health, is no exception. These nanoparticles have been designed from biocompatible and biodegradable polymers, a characteristic that confers to them a certain level of reliability when used in terms of health by demonstrating effects as coadjuvant antimicrobial agents [16], anticancer [17], and antioxidants [18]. The polymeric nanoparticles’ potential in medicine has been experimented with as influencers of the immune system directly by their composition or indirectly as carriers of active substances [19]. In vitro systems for drug delivery based on nanocarriers have been developed to load the antioxidant drug curcumin on biocompatible PCL nanoparticles (Cur-PCL/F68 NPs), evaluating their in vitro cell viability in neuronal (SH-SY5Y) and glial (MIO-M1) cells as an approach to examine the behavior of NPs in biological environments [20]. In polymeric nanoparticle synthesis, it has been possible to confer the property of integrating bioactive molecules and protecting them from degradation in their physiological distribution pathway to meet their objective of reaching the critical area of action [14].

Several investigations have highlighted the zein molecule’s versatility. zein is a prolamin protein found in corn, is insoluble in water, and can form complex nanostructures to encapsulate different compounds. Zein-based films, for example, zein conjugate with essential oils of garlic and thyme (where the oil mixture was added at 0%, 2%, 3%, and 5% (*v*/*v*) to the zein films) showed inhibitory activity against all tested bacteria (*enteropathogenic Escherichia coli* (EPEC), *Listeria monocytogenes*, *Salmonella Enteritidis*, and *Staphylococcus aureus*) related to food contamination, with inhibition halos of between 6.5 mm and 8.27 mm. Furthermore, the results showed that the coating could be applied as a carrier to increase the shelf life of food products [21]. Similarly, flavonoids with glycoside groups on the A-ring exhibited excellent zein-binding ability, and their binding process to zein was driven by a hydrophobic force [22]. In another investigation, tocopherol was successfully encapsulated in a zein–chitosan complex; the results indicated encapsulation of tocopherol, and the particle size and ζ-potential of the complex ranged from 200 to 800 nm and from +22.8 mV to +40.9 mV, respectively. Compared to zein nanoparticles, the zein–chitosan complex provided more significant control of tocopherol release under simulated gastrointestinal tract conditions [23]. Another investigation studied the potential of a zein nanoparticle-based system for the oral administration of resveratrol in humans and its impact on pharmacokinetics. These nanoparticles showed a mean size of 331 nm with an encapsulation efficiency of 87%. A single dose of 250 mg of resveratrol was administered to 16 healthy volunteers. The concentrations of resveratrol and its metabolite resveratrol-3-0-*D*-glucuronide were monitored to find the relationship between the amount and these parameters [24].

Similarly, quercetin and vitamin A were encapsulated in zein nanoparticles, and the results showed that the encapsulation efficiency depends on their initial concentration in the synthesis mixture. In the presence of an initial 0.05 g of quercetin, the encapsulation efficiency was 37%. The encapsulation efficiency for 50 µL of vitamin A was 44%. Furthermore, with the increase in zein concentration and the addition of sodium caseinate as a stabilizer, the encapsulation efficiency increased from 44% to 85% [25].

Biopolymer composite nanoparticles based on zein may be an effective oral delivery system for flavonoids such as quercetin; it could provide alternative antioxidant activity of quercetin delivered in the form of nanoparticles and used in pharmaceutical formulations [26]. Another attractive composite is epigallocatechin-3-gallate-loaded zein nanoparticles. It has been used for its biological properties related to skin pigmentation and sun protection [27]. Otherwise, a green reductant like an aqueous extract of *Citrus sinensis* in Ag nanoparticle synthesis can create a capping without any toxic reagent in the formulation. These nanoparticles revealed excellent antioxidant and anti-human-lung-cancer effects and may be used in medicine as a chemotherapeutic supplement or drug [28]. There have been investigations into the beneficial applications of nanoparticles. However, there is evidence of great endeavors and the necessity to optimize the size, shape, stability, delivery, and other characteristics. Mainly, there is a necessity to evaluate the potential risks of nanoparticles before their applications in clinical practice in cancer treatment [29].

In the present research work, the antioxidant effect of nanoparticles based on zein and orange extract (*Citrus sinensis*) was evaluated, and the extract was obtained by ultrasound-assisted extraction. Subsequently, physicochemical characterization was carried out, and the antioxidant capacity was evaluated.

## 2. Materials and Methods

### 2.1. Ultrasonic-Assisted Extraction of Bioactive Compounds from Citrus Peel

*Citrus sinensis* Valencia oranges obtained from the local market were used. First, the peel was washed, cut into eighths, and dried in a convection oven (Lindberg/Blue Model MO1490A-1, Asheville, NC, USA) for 24 h at 45 °C. It was then ground in a food processor (Nutribullet 600 W) and sorted to 0.85 mm using a T20 sieve obtained from FICC S.A. de CV, Mexico. The next step was to take 2.5 g of the sample and place it in a beaker containing 10 mL of water. The extraction was performed in a Qsonica Model Q500 ultrasonic cleaner at a frequency of 20 kHz, amplitude of 30%, and two periods of 10 min with 1 min of rest. Subsequently, the extract was filtered in a filtration kit and centrifuged at 4 °C and 30 min at 6000 rpm in an OHAUS centrifuge, Model FC571812. Finally, the filtered extract was lyophilized (Labconco FreeZone 12, Kansas City, MO, USA) at −61 °C, 0.00024 Pa for 72 h.

### 2.2. Synthesis of Zein Nanoparticles

Zein nanoparticles (NpZ) were prepared using a modified method based on the antisolvent procedure [30]. First, 142.85 mg of zein (CAS 9010-66-6 Sigma-Aldrich, St. Louis, MI, USA) was dissolved in 10 mL of ethanol/water binary solvent (90:10 *v*/*v*) to form a stock solution. Then, 2 mL of the stock solution was added to 9 mL of deionized water using a peristaltic pump with a constant flow rate of 0.6 mL/min. This process was performed under continuous magnetic stirring (1000 rpm) (Figure 1). The samples were stored at 4 °C for subsequent physicochemical characterization (DLS, ζ-potential, and SEM). The final dispersions were freeze-dried for 24 h to obtain solid nanoparticle powder samples. The powder samples were used to evaluate their antioxidant activity.

### 2.3. Physicochemical Characterization of Nanoparticles

A Malvern Zetasizer NanoSeries was used to measure the size distribution and ζ-potential of the samples. In addition, SEM (JEOL JSM-7800F, Pleasanton, CA, USA) was used to evaluate the morphology of the nanoparticles. Samples were diluted 1 to 10 before deposition on an aluminum film. FTIR spectra were collected on a Spectrum spectrometer (Perkin Elmer Inc., Waltham, MA, USA) equipped with a single attenuated total reflectance (ATR) diamond set to a range of 400 cm^−1^ to 4000 cm^−1^ [31].

### 2.4. Quantification of Total Phenol Content (NpZ, NpZOE, and OE)

An incubation process was carried out using test tubes with an amount of 50 µL of each sample incubated with 3.0 mL of distilled water and 250 µL of Folin–Ciocalteu reagent, 750 μL of 20% Na_2_CO_3_, and 950 μL of distilled water. After 3 min, 1.0 mL of a sodium carbonate solution was added and allowed to react for 60 min in the dark [32]. A gallic acid curve was prepared, and the results are expressed as gallic acid equivalents per gram of dry matter (mg GGE/g d.w.). All analyses were performed in triplicate.

### 2.5. Determination of Total Flavonoids (NpZ, NpZOE, and OE)

The aluminum chloride (AlCl_3_) assay was used [33]. Each sample (0.5 mL) was mixed with 2 mL of distilled water and 0.15 mL of 10% (*w*/*v*) AlCl_3_. After 10 min of incubation in the dark at room temperature, 1 mL of 1 M sodium hydroxide (NaOH) and 1.2 mL of distilled water were added to the mixture. After 15 min incubation in the dark at room temperature, the mixture was measured in a microplate reader (Veloskan™LUX, Thermo Fisher Scientific, Waltham, MA, USA) at 430 nm. The standard used was quercetin. Therefore, results are given as milligrams of quercetin equivalent per gram of dry weight (mg QE/g d.w.) [34].

### 2.6. Antioxidant Capacity ABTS Assay (NpZ, NpZOE, and OE)

The method consisted of reducing the green/blue coloration produced by the reaction of ABTS with the antioxidant present in the sample. To generate a cation radical, 19.2 mg of ABTS was dissolved in 5 mL distilled water. The ABTS radical was spread by oxidation of ABTS (7.0 mM) with potassium persulfate (K_2_S_2_O_8_, 4.95 mM) in the dark (12 h) at 25 °C. Next, ABTS was diluted with PBS (0.2 M, pH 7.4) to reach an absorbance of 0.7 measured at 734 nm. Finally, 20 mL of sample and 200 mL of ABTS solution were taken and allowed to react for 30 min, and the absorbance was measured at 734 nm in a microplate reader (Veloskan™ LUX, Thermo Fisher Scientific, Waltham, MA, USA). The trapping activity is reported as ABTS radical inhibition (%) [35]. Equation (1) was used to determine the scavenging capacity.
(1)$$Scavenging\;capacity=\left(1-\left(\frac{Abs_{1}-Abs_{2}}{Abs_{0}}\right)\right)\times 100$$
where *Abs*_0_ is the control absorbance (water), *Abs*_1_ is the absorbance of the sample with ABTS, and *Abs*_2_ is the absorbance of the water with ABTS.

### 2.7. FRAP Assay (NpZ, NpZOE, and OE)

The ferric reduction ability was evaluated based on the FRAP assay. FRAP reagent was prepared in acetate buffer (pH 3.6), 10 mM, 2,4,6-tri(2-pyridyl)-s-triazine (TPTZ) solution in 40 mM HCl, and 20 mM iron (III) chloride solution in 10:1:1 (*v*/*v*/*v*) ratios [35]. The treatments were diluted in different concentrations. A volume of 20 µL of the sample was taken, and 280 µL of FRAP reagent was added. As a reference, a solution of Trolox standard was prepared. Then, the reaction was carried out for 30 min, and the absorbance was measured at 638 nm in a microplate reader (Veloskan™ LUX, Thermo Fisher Scientific, Waltham, MA, USA). The results are expressed as micromoles of Trolox equivalent per gram of dry sample (µmol TE/g d.w.).

### 2.8. DPPH Radical Scavenging Assay for NpZ, NpZOE, and OE

DPPH radical scavenging assays were performed according to Hu et al., 2015 [36]. First, the treatments were diluted in different concentrations. Next, each concentration (50 µL) and methanolic DPPH solution (200 µL) at a concentration of 0.4 mM were added to a 96-well microplate. Then, the reaction was carried out in the dark at 25 °C for 30 min. After that, the absorbance was measured at 515 nm using a microplate reader (Veloskan™LUX, Thermo Fisher Scientific, Waltham, MA, USA). Finally, a Trolox standard solution was prepared and assayed under the same conditions [35]. Additionally, the antioxidant concentration corresponding to 50% inhibition of the DPPH radical (EC_50_ DPPH) was calculated from the graph of the scavenging percentage versus the concentration of the antioxidant (mg/mL) tested using linear regression [32].

## 3. Results and Discussion

### 3.1. Physicochemical Characterization

Figure 2 shows the SEM results of the NpZ and NpZOE samples, respectively. The micrographs show a well-defined spherical morphology. The addition of EO resulted in less agglomeration in the NpZOE sample compared to NpZ, which is related to less coalescence and, thus, higher stability. Likewise, the standard deviation of the NpZOE sample decreased relative to that of NpZ. This suggests greater size control caused by adding the extract. These results agree with the DLS measurements, in which a decrease in polydispersity is also observed (Table 1). This effect can also be observed in the size distribution plots obtained from SEM micrographs, where a significant decrease in size populations can be seen (Figure 3).The decrease in ζ-potential (Table 1) is attributed to the presence of OE on the particle surface. This change in surface charge is due to the hydroxyl groups contained in the OE.

This behavior was previously reported when studying resveratrol-loaded zein particles coated with alginate, where a decrease in the zeta potential was observed with increasing alginate concentration and increasing number of negative charges present on the surface [37]. The results show that zein nanoparticles with orange extract with a positive surface charge were obtained. This electrical character can promote cell uptake due to its electrostatic interaction between the nanoparticle surface and cell membrane [38]. The particle sizes observed in the SEM micrographs are 86.5 ± 34.6 nm and 150.1 ± 31.8 nm for NpZ and NpZOE, respectively. The size of NpZ is smaller than that reported by Luo et al. [23], who reported zein particles with a diameter up to 364 nm. This can be attributed to the concentration of zein used previously, 5 mg/mL, which is much higher than that used in the present investigation (0.5 mg/mL). The hydrodynamic diameter observed in DLS shows an increase in size caused by the presence of OE during the nanoparticle formation process. The preceding coincides with the effect observed in the SEM micrographs. Previous studies reporting the encapsulation of natural extracts show an increase in particle size upon addition of the extract, caused by the spatial arrangement of the amino acid chains of zein interacting with the encapsulated extract molecules [24].

### 3.2. FT–IR Spectra

Figure 4 shows the OE spectrum; an intense signal is observed at 3231 cm^−1^ due to the stretching of the O-H bond present in the phenolic and alcoholic groups of flavonoids and organic acids. A prominent band is seen at 1594 cm^−1^ due to the stretching of the C=C bond present in aromatic rings. One more signal around 1404 cm^−1^ may be associated with stretching of the C=O bond. A band is observed from 1308 cm^−1^ to 1232 cm^−1^ due to hydrogen bonds [27]. Finally, a signal is observed at 1068 cm^−1^, indicating the presence of C-O-C stretching of ester groups [31].

In the case of NpZ, the characteristic bands corresponding to amides I, II, and III are 1651 cm^−1^ due to C=O stretching, 1537 cm^−1^ due to N-H bending, and 1242 cm^−1^ due to C-N stretching [39]. In addition, a band is detected at 3299 cm^−1^ due to the OH groups’ axial stretching and the N-H bond. Compared with that of NpZ, in the spectrum of NpZOE, the signal of 3299 cm^−1^ was shifted to 3287 cm^−1^; this migration of the characteristic absorption band meant that the interaction of OE and zein affected the formation of a hydrogen bond between zein groups and flavonoid groups (OH and C=O). In addition, zein’s characteristic amide I and II bands were located at 1651 cm^−1^ and 1537 cm^−1^, respectively. Upon the inclusion of OE, the absorption peak of the amide I band shifted to 1645 cm^−1^, while the amide II band shifted to 1516 cm^−1^. Previous results reported this effect and attributed the signal shift to the formation of hydrogen bridges and hydrophobic interactions. The above-mentioned elucidates a stable system’s formation through non-covalent interactions [27,40].

### 3.3. Quantification of Total Phenols and Flavonoids

Table 2 shows total phenols of 132.31 ± 5.14 µmol Eq AG/g dry weight. This result is higher than that reported by other authors, such as Rodsamran et al. [41], who obtained 54 ± 1 µmol Eq AG/g ps under the conditions of amplitude 30%, time 4 min, and 50% ethanol in lime peel, as well as Londoño et al. [42], who obtained 19.5 µmol Eq AG/g ps at 60 kHz over 30 min in water (1:10) in orange peel. Dahmoune et al. [43] reported 13.5 µmol Eq AG/g ps with an amplitude of 65.94% at 27 kHz for 8.33 min at 2 °C using 75.79% aqueous acetone. On the other hand, NpZ presented significantly higher total phenols than did the OE, with a value of 311.02 ± 8.20 µmol Eq AG/g; this is because zein, by its protein nature, belongs to the prolamins; therefore, it presents in its structure predominantly proline [44], an aromatic amino acid detected by Folin–Ciocalteu’s reagent in the technique [45]. This decrease in the value of total phenols compared to NpZ is because phenols are a secondary metabolite present in citrus/orange peel which is found in minimal amounts [46], while in pure zein, proline is predominant [44], even though the ratio between zein and extract favors the extract.

Table 2 presents the quantification of flavonoids referring to quercetin as an antioxidant model (µmol Eq QC/g dry weight). The OE presented 6.73 ± 0.62 and NpZ presented 30.47 ± 1.49, while NpZOE presented a higher amount of 46.50 ± 1.10 µmol Eq QC/g dry weight. Campo et al., 2016 obtained from 24 µmol to 36 µmol Eq QC/p of orange peel with a UAE pretreatment at 45 kHz at a temperature of 30 °C for 30 min. Londoño et al., 2010 obtained 40.25 ± 12.09 µmol Eq QC/p at 60 kHz over 30 min at 40 °C using water (1:10). The Eq value of quercetin is lower than that reported by other authors, possibly because the conditions used in this study were milder; only water at room temperature was used, while the other studies used organic solvents in their extraction. In addition, it was observed that the same behavior occurred in the quantification of flavonoids as in the total phenolics technique concerning the zein nanoparticles and the complex.

The different samples present phenolic compounds responsible for inhibiting the activity of free radicals and reactive oxygen species. Previously, the possible mechanism of action of phenols was reported: they can act by interrupting the lipid oxidation reaction, inhibiting chemiluminescence reactions by capturing several reactive oxygen species [47]. In addition, the encapsulation of these extracts favors stability and protection against external factors such as contact with oxygen, temperature, and light, which cause their degradation [27,48].

### 3.4. Antioxidant Capacity—ABTS Assay

Figure 5 shows each treatment’s ABTS radical inhibition capacity at the different concentrations tested. The activity levels of OE and NpZOE were statistically equivalent to that of Trolox at the maximum concentration tested (500 µg/mL). On the other hand, no significant difference was found between OE and NpZOE at 500 µg/mL, 250 µg/mL, and 31.3 µg/mL. The results indicate that the NpZOE treatment at all concentrations tested presented higher antioxidant activity than the NpZ treatment. Similar results have been reported regarding an increase in the antioxidant capacity of zein nanoparticles when including bioactive molecules that improve the ABTS radical trapping capacity, for example, zein–turmeric and zein–quercetin nanoparticles compared to empty zein nanoparticles [49]. This effect can mainly be attributed to the higher concentration of OE included in the NpZOE formulation.

A statistically significant result (*p* < 0.05) is that higher antioxidant activity was observed at 7.8 µg/mL for the NpZOE treatment concerning NpZ, OE, and Trolox. Even at this concentration, antioxidant capacity above 10% was observed for the NpZOE treatment. This effect may be due to some antioxidant activity associated with the differences in concentration of bioactive molecules in OE, NpZ, and NpZOE, with 6.73 ± 0.62 µmol, 30.47 ± 1.49 µmol, and 46.50 ± 1.10 µmol Eq QC/g d.w., respectively (Table 2). It has been reported that coupling bioactive molecules with biopolymeric nanoparticles favors the stability and bioavailability of these compounds [48,50].

### 3.5. Antioxidant Capacity—DPPH Assay

The DPPH test is widely used to evaluate free radicals’ scavenging ability and total antioxidant capacity. Figure 6 shows the DPPH radical stabilization activity. The NpZOE and OE treatments were statistically equivalent, showing DPPH radical inhibition capacity higher than 35% at the 500 µg/mL concentration, while NpZ showed lower activity at the same concentration. However, at the 250 µg/mL concentration, the NpZOE treatment showed a higher antioxidant capacity, above 13% inhibition of the DPPH radical. At lower concentrations, the same behavior was observed, i.e., NpZOE treatment was more effective than NpZ and OE. This behavior has also been reported in thymol-loaded zein nanoparticles, which reached 32% inhibition of the DPPH radical, and the inhibition increased with increasing concentration of thymol in the zein nanoparticles [30]. Similar results have been published in a determination of the antioxidant capacity of lime peel extracts extracted by EUA (amplitude 30%, time 4 min, and ethanol 50%), showing total phenols of 450 µEq AG/g d.w. and antioxidant capacity for DPPH of 11 µEq Trolox/d.w. [39].

### 3.6. IC_50_ Concentration 

In the DPPH and ABTS assays, IC_50_ is the antioxidant concentration required to obtain 50% radical inhibition [50]. It is crucial to determine the antioxidant capacity of NpZ, NpZOE, and the OE bioactive antioxidant to help select the most favorable conditions ensuring the extraction of natural antioxidants with beneficial effects on health. Therefore, their concentrations responsible for 50% radical scavenging activity against 350 M of the DPPH and ABTS radicals were determined. The IC_50_ values for OE scavenging DPPH and ABTS radicals were 49.86 µg/mL and 572.5 µg/mL, respectively (Table 3). Previous research reported IC_50_ values for zein and zein–epigallocatechin-3-gallate nanoparticles of 996 µg/mL and 113 µg/mL, respectively, for nanoparticles with 285.8 nm and 340 nm diameters and ζ-potential values of 29.6 ± 0.8 mV and 37.9 ± 0.32 mV [37]. Therefore, the treatments quantified the DPPH and ABTS radical scavenging activity in terms of the inhibition percentage of the free radical by the treatments and the IC_50_ values in µL/mL. For ABTS, NpZ presented IC_50_ 391.03 µg/mL, which is higher than that of NpZOE, indicating the effectiveness of integrating orange extract into zein nanoparticles.

Table 4 presents the antioxidant capacity results of the tested treatments, expressed in Trolox equivalent micromoles per gram of dry sample (μmol TE/g d.w.). The ABTS of the orange extract incorporated into the zein nanoparticles was approximately 863.17 µM TE/g d.w., which means that the antioxidant capacity of 1 g of NpZOE was equivalent to that of 863.17 µM Trolox [49]. Higher Trolox equivalent antioxidant capacity (TEAC) values represent higher antioxidant activity, indicating greater capacity to scavenge free radicals by hydrogen transfer and to reduce Fe(III) to Fe(II) [47]. The DPPH and ABTS assay results show that NpZOE presented higher antioxidant activity than did OE and NpZ (*p* < 0.05). Similar results have been published; the TEAC was compared between curcumin solubilized in ethanol and curcumin encapsulated into zein–pectin nanoparticles. Encapsulated curcumin was about twice as efficient at scavenging ABTS as curcumin solubilized in ethanol, and empty (curcumin-free) nanoparticles exhibited feeble ABTS scavenging capacity [49]. The determination of the antioxidant capacity by FRAP indicated that the orange extract (OE) presented greater capacity (*p* < 0.05) in terms of Fe(III) to Fe(II) reduction, which could be attributed to the metal chelating capacity of the molecules contained in the extract. Contrary to previous research, the results show that both the ABTS scavenging capacity and ferric-ion-reducing power of quercetin–zein-loaded nanoparticles were significantly higher than those of quercetin in solution [26].

## 4. Conclusions

UAE in water is an effective method of obtaining the phenolic compounds present in orange peels, and the nanoprecipitation method under the conditions used allowed for obtaining particles in the nanometric range with positive ζ-potential. The interaction studies show the formation of hydrogen bonds and hydrophobic interactions between zein and orange extract and elucidate a stable system’s formation through noncovalent interactions. These characteristics are desirable in the area of cellular internalization of bioactive molecules. The antioxidant capacity analysis indicated a high antioxidant capacity of both orange extract and the zein nanoparticles with orange extract. As stated above, this opens up potential applications in the design of systems for treating pathologies caused by excess oxidizing species. The results suggest that two different mechanisms may determine the antioxidant activity of NpZOE, donating hydrogen atoms (DPPH) or electrons (ABTS), thus protecting against oxidative damage and free radical stabilization. This study presents the possibility of obtaining orange extracts by ultrasound and coupling them to zein-based nanoparticulate systems to be applied as biomedical materials functionalized with antioxidant substances of pharmaceutical utility.

## Figures and Tables

**Figure 1 materials-15-04838-f001:**
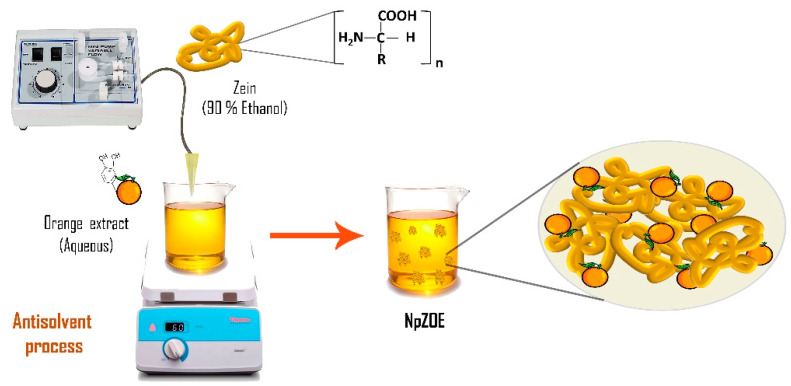
Schematic representation of the synthesis of zein-loaded orange extract nanoparticles.

**Figure 2 materials-15-04838-f002:**
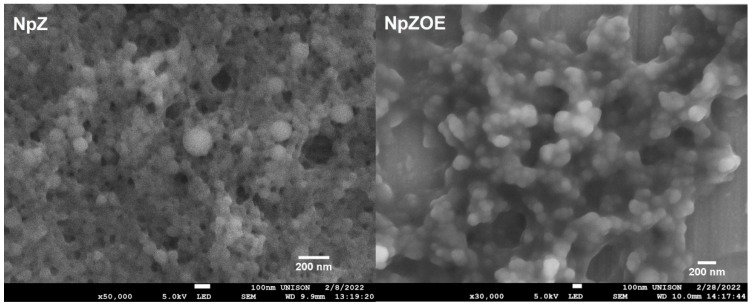
SEM micrographs of zein nanoparticles (NpZ) and zein nanoparticles with orange extract (NpZOE).

**Figure 3 materials-15-04838-f003:**
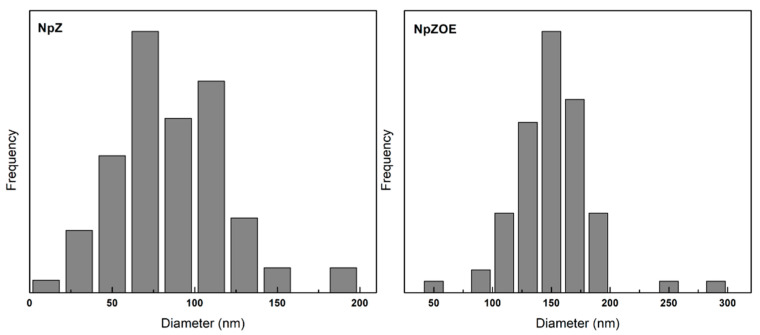
Particle size dispersion (nm) of zein nanoparticles (NpZ) and zein nanoparticles with orange extract (NpZOE).

**Figure 4 materials-15-04838-f004:**
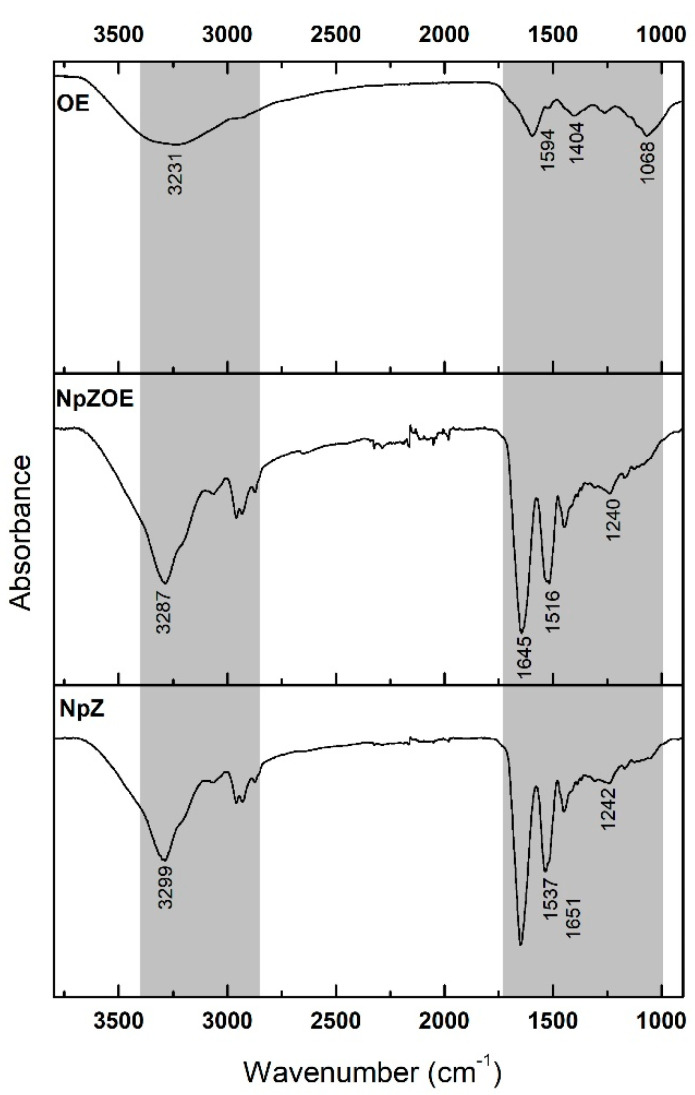
Infrared spectra of orange extract (OE), zein nanoparticles (NpZ), and zein nanoparticles with orange extract (NpZOE). The shaded gray areas represent the ranges in which the tracked signals are located.

**Figure 5 materials-15-04838-f005:**
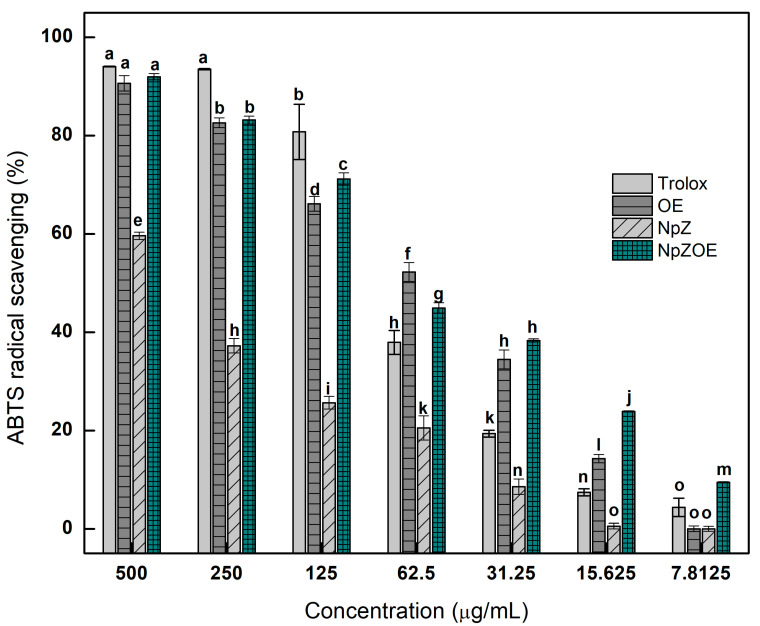
Antioxidant capacity of orange peel extract (OE), zein nanoparticles (NpZ), and zein nanoparticles with orange extract (NpZOE) using Trolox as a reference antioxidant. ^a–o^ Different literals indicate a significant difference (*p* < 0.05).

**Figure 6 materials-15-04838-f006:**
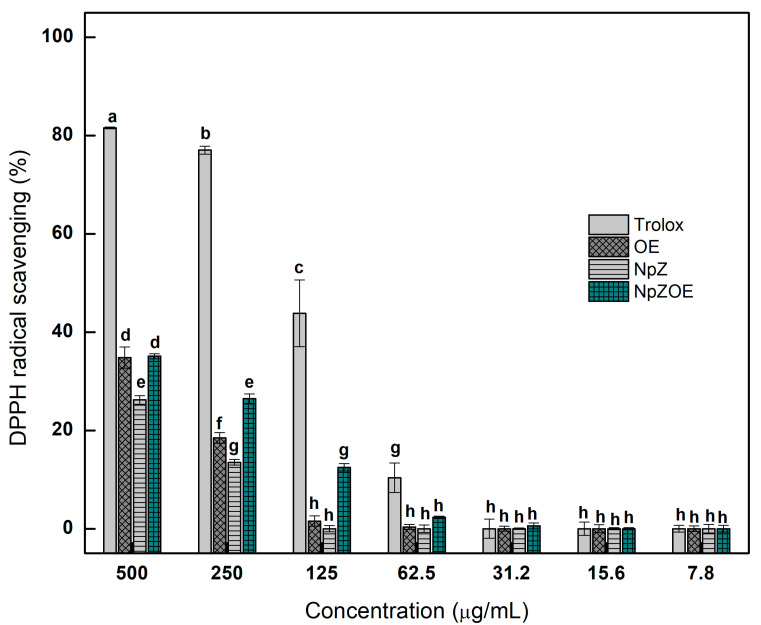
Antioxidant capacity of orange peel extract (OE), zein nanoparticles (NpZ), and zein nanoparticles with orange extract (NpZOE) by DPPH using Trolox as a reference antioxidant. ^a–h,^ Different literals indicate a significant difference (*p* < 0.05).

**Table 1 materials-15-04838-t001:** Hydrodynamic diameter (nm) and zeta potential (mV) values of zein nanoparticles (NpZ) and zein nanoparticles with orange extract (NpZOE).

	NpZ	NpZOE
Hydrodynamic diameter (nm)	159.26 ± 5.96 ^b^	199.96 ± 2.87 ^a^
ζ-potential (mV)	+ 22.63 ± 1.52 ^b^	+ 11.86 ± 0.63 ^a^
PDI	0.182	0.058

^a,b^ Different literals per row indicate a significant difference (*p* < 0.05).

**Table 2 materials-15-04838-t002:** Concentrations of flavonoids and total phenols determined in orange extract (OE), zein nanoparticle (NpZ), and zein nanoparticle with orange extract (NpZOE) samples.

	Total Phenols µmol Eq AG/g Dry Weight	Flavonoids µmol Eq QC/g Dry Weight
OE	132.31 ± 5.14 ^a^	6.73 ± 0.62 ^a^
NpZ	311.02 ± 8.20 ^b^	30.47 ± 1.49 ^b^
NpZOE	254.02 ± 13.40 ^c^	46.50 ± 1.10 ^c^

^a,b,c^ Different literals per row indicate a significant difference (*p* < 0.05).

**Table 3 materials-15-04838-t003:** IC_50_ of OE, NpZ, and NpZOE samples, according to the ABTS and DPPH determination methods.

	IC_50_ (µg/mL) ABTS	IC_50_ (µg/mL) DPPH
OE	49.86	572.50
NpZ	391.03	879.81
NpZOE	87.41	652.32

The antiradical activity is expressed as IC_50_ (µg/mL), the concentration required to cause 50% ABTS and DPPH inhibition under the defined conditions.

**Table 4 materials-15-04838-t004:** Antioxidant capacity of orange extract (OE), zein nanoparticles (NpZ), and zein nanoparticles with orange extract (NpZOE) determined by ABTS, DPPH, and FRAP. The results are reported as Trolox equivalent micromoles per gram of dry sample (μmol TE/g d.w.).

	ABTS	DPPH	FRAP
OE	851.78 ± 2.79 ^a^	1604.83 ± 15.21 ^a^	2506.65 ± 9.71 ^a^
NpZ	547.13 ± 8.71 ^b^	1429.61 ± 10.47 ^b^	2239.27 ± 115.15 ^b^
NpZOE	863.17 ± 6.29 ^c^	1614.90 ± 9.71 ^c^	2269.49 ± 20.99 ^b^

The data are represented as mean values ± standard deviation (n = 3). Different letters within the same row indicate statistically significant differences (*p* < 0.05).

## Data Availability

The data presented in this study are available on request from the corresponding author.

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
