# Peer review of "Antioxidant Effect of Nanoparticles Composed of Zein and Orange (Citrus sinensis) Extract Obtained by Ultrasound-Assisted Extraction"

_materials, 2022, doi:10.3390/ma15144838_

Round 1

Reviewer 1 Report

In the present manuscript, the authors reported ntioxidant effect of nanoparticles composed of Zein and or-2ange extract obtained by ultrasound-assisted 3extraction. It may bring a valuable reference to the researchers in this area. The paper could be interesting and attractive for the readership of the Materials journal. Publication is commended in present form.

Reviewer 2 Report

The manuscript entitled Antioxidant effect of nanoparticles composed of Zein and orange (Citrus Sinensis) extract obtained by ultrasound-assisted extraction describes the extraction of orange extract and its encapsulation in zein matrix for the physicochemical characterization. It also evaluates the antioxidant capacity of the encapsulated orange extracts in zein matrix. Add more information about the synthesis of nanoparticle composed of Zein and orange extract. Also add more information in the introduction and conclusion part. Explain with more details the results XRD analysis obtained . The paper is interesting, presents new insights, and is scientifically precise in details. Presenting results, as well as figures, are of good quality. This paper could be published in materials. 

Reviewer 3 Report

The authors demonstrated the synthesis, characterization and antioxidant property of orange extract@Zein nanoparticles. This research work deserved to publish in this journal without any further revision. 

The authors demonstrated the ability of zein nanoparticles for the encapsulation of orange extract; the composite thus synthesized namely NpOZ was well characterized by using several techniques such as FT-IR, DLS – Zeta potential, SEM etc., which revealed the host-guest interaction between Zein and orange extract. More interestingly, this research paper presents insights to the design of novel nanomaterials as antioxidants for pharmaceutical applications. The manuscript is well written and organized in a good way. Although this paper can be accepted in its current form, the following comments will be helpful for the further improvement of the manuscript.

1.       Give a schematic figure for the design and synthesis of NpOZ

2.       Improve the introduction section: Highlight the challenges and importance of nanoparticle-based antioxidants in the pharmaceutical industry with relevant references.

3.       Discussion section can be improved by taking relevant examples from the literature, and highlighting the impact of the NpOZ.

4.       The scale bar of the FE-SEM image (Figure 1) needs to be more visible.

5.       Powder XRD comparison data of zein nanoparticles, NpOZ is missing. 
